# Application of Chlorophyll Fluorescence Analysis Technique in Studying the Response of Lettuce (*Lactuca sativa* L.) to Cadmium Stress

**DOI:** 10.3390/s24051501

**Published:** 2024-02-26

**Authors:** Lina Zhou, Leijinyu Zhou, Hongbo Wu, Tingting Jing, Tianhao Li, Jinsheng Li, Lijuan Kong, Fengwu Zhu

**Affiliations:** College of Engineering and Technology, Jilin Agricultural University, Changchun 130118, China; zhoulina@jlau.edu.cn (L.Z.); zhouleijinyu8514@163.com (L.Z.); whb05082023@163.com (H.W.); 13274422035@163.com (T.J.); 18131571123@163.com (T.L.); jinshengl@jlau.edu.cn (J.L.); konglijuan630@sina.com (L.K.)

**Keywords:** *Lactuca sativa* L., cadmium stress, estimation model, relative chlorophyll content, chlorophyll fluorescence parameters

## Abstract

To reveal the impact of cadmium stress on the physiological mechanism of lettuce, simultaneous determination and correlation analyses of chlorophyll content and photosynthetic function were conducted using lettuce seedlings as the research subject. The changes in relative chlorophyll content, rapid chlorophyll fluorescence induction kinetics curve, and related chlorophyll fluorescence parameters of lettuce seedling leaves under cadmium stress were detected and analyzed. Furthermore, a model for estimating relative chlorophyll content was established. The results showed that cadmium stress at 1 mg/kg and 5 mg/kg had a promoting effect on the relative chlorophyll content, while cadmium stress at 10 mg/kg and 20 mg/kg had an inhibitory effect on the relative chlorophyll content. Moreover, with the extension of time, the inhibitory effect became more pronounced. Cadmium stress affects both the donor and acceptor sides of photosystem II in lettuce seedling leaves, damaging the electron transfer chain and reducing energy transfer in the photosynthetic system. It also inhibits water photolysis and decreases electron transfer efficiency, leading to a decline in photosynthesis. However, lettuce seedling leaves can mitigate photosystem II damage caused by cadmium stress through increased thermal dissipation. The model established based on the energy captured by a reaction center for electron transfer can effectively estimate the relative chlorophyll content of leaves. This study demonstrates that chlorophyll fluorescence techniques have great potential in elucidating the physiological mechanism of cadmium stress in lettuce, as well as in achieving synchronized determination and correlation analyses of chlorophyll content and photosynthetic function.

## 1. Introduction

With the rapid progress of industrialization and urbanization in China, the heavy metal pollution of soil and water caused by industrial and agricultural waste, as well as household garbage, is showing a trend of continuous aggravation. In the process of greenhouse cultivation, the application of pesticides and fertilizers is considered to be the main cause of heavy metal accumulation in soil [1,2]. Among them, cadmium, a heavy metal with teratogenic, carcinogenic, mutagenic, and neurotoxic effects, has been listed as the primary pollutant in contaminated soil areas [3,4,5]. Cadmium is a non-essential element for plant growth and is highly toxic. It is easily absorbed and enriched by plants and can enter the human body through the food chain of “soil–plant–food”. After accumulating in the human body, it poses a threat to human health. Plants that are subjected to long-term cadmium stress experience impaired normal growth and development. Cadmium stress can inhibit photosynthesis in plants, leading to reduced absorption of water and nutrients. As a result, plants may exhibit slow growth, poor leaf development, wilting, chlorosis, and even death [6,7,8]. Lettuce has a high nutritional value and is known for its various health benefits. It has cooling and calming properties, promotes liver and gallbladder health, and helps to lower cholesterol levels [9]. Moreover, lettuce is highly sensitive to the environment and has the ability to accumulate heavy metals, particularly in its stems and leaves. Therefore, lettuce is often used as a test material in studying soil cadmium pollution [10].

During the process of photosynthesis, most of the absorbed light energy by plants is used for photochemical reactions. The remaining energy is dissipated as heat or emitted as fluorescence [11,12,13]. Although the chlorophyll fluorescence signal accounts for only 3–5% of the total absorbed energy, it provides sufficient diagnostic evidence for the growth status of plants under stress conditions [14]. When plants are subjected to stress, the photosynthetic capacity of their leaves undergoes changes. The chlorophyll fluorescence technique, known as a rapid and non-destructive probe of plant chlorophyll, is a primary method for studying the functional aspects of photosynthetic machinery in plants under stress. It effectively reflects the ability of plant leaves to absorb and utilize light energy under adverse conditions [15]. Gan et al. [16] applied the chlorophyll fluorescence technique to investigate the Distylium chinense cadmium pollution. The results indicated that chlorophyll fluorescence parameters could provide a physiological explanation for the plant’s tolerance. Under cadmium stress, the photosystem II (PSII) in plants suffered reversible damage in the light reaction center. Wang et al. [17] utilized the chlorophyll fluorescence technique to analyze the combined effect of cadmium and fluoride on lettuce. The results demonstrated that lettuce responded to the pollutants by reducing the quantum yield of PSII and the electron transfer activity. Wan et al. [18] studied the toxic mechanism of cadmium on the photosynthetic system of poplar using chlorophyll fluorescence technology. The results showed a significant decrease in the photosynthesis and the quantum efficiency of the primary photochemical reaction of PSII under cadmium stress.

The above demonstrates that the chlorophyll fluorescence technique has been widely applied and promoted in the fields of photosynthesis and plant stress physiology. Therefore, in order to explore the influence of the heavy metal cadmium on the relative chlorophyll content (SPAD value) and the fluorescence kinetic characteristics of lettuce seedlings in the environment, this study used lettuce seedlings as the experimental material and applied the chlorophyll fluorescence analysis technique. The main focus of this study was to investigate the changes in SPAD values, rapid chlorophyll fluorescence induction kinetics curve, and related chlorophyll fluorescence parameters of lettuce leaves under cadmium stress. Additionally, an estimation model for SPAD values of lettuce leaves was established. The aim was to uncover the mechanisms through which cadmium stress, a heavy metal, affects photosynthesis in plants and achieves synchronized determination and correlation analyses of chlorophyll content and photosynthetic function.

## 2. Materials and Methods

### 2.1. Materials and Experimental Design

This experiment was conducted in May 2023 within the campus of Jilin Agricultural University in Changchun, Jilin Province (125°42′ E, 43°82′ N). Potted planting methods were used with Great Speed Lettuce (*Lactuca sativa* L. cv. Grand Rapids, purchased from Kui Shou Agricultural Technology Co., Ltd., Langfang, China) as the test material. The experimental soil used was uncontaminated nutrient soil (purchased from Zhongnong Saishi Agricultural Technology Co., Ltd., Changchun, China), with a high content of elemental nitrogen, phosphorus, and potassium at 3.72%, and an organic matter content of 32.9%, both calculated on a dry weight basis. Impurities were removed from the soil using a sieve, making it fine-grained, and then the soil was placed in a dry, ventilated area to settle for 3 days. The cadmium content in the soil was set at 5 concentration gradients, namely 0 (CK), 1, 5, 10, and 20 mg/kg. Distilled water was used as the solvent and cadmium nitrate as the exogenous cadmium additive to prepare a 200 mL solution. The prepared cadmium solutions of different concentrations were sprayed layer by layer onto the corresponding test soil and thoroughly mixed. After aging for 7 days, the soil was transferred into flower pots with an outer diameter of 17 cm and a depth of 15 cm, with 1 kg of soil in each pot [19]. Four replicates were set for each treatment level, and lettuce seedlings with consistent and vigorous growth were selected for transplantation into the flower pots, with one plant being transplanted into each pot. To ensure the normal growth of lettuce seedlings throughout the entire experiment, ample water supply was ensured. The plants were watered daily, keeping the soil moisture at around 70%. After cadmium stress, the SPAD values and chlorophyll fluorescence parameters of lettuce seedling leaves were measured on the 1st, 3rd, 5th, 7th, and 9th day for the five different treatments. On the 9th day after cadmium stress, the rapid chlorophyll fluorescence induction kinetics curves (OJIP curve) of lettuce seedling leaves for the five different treatments were measured. At this time, the first leaf had unfolded five leaves. 

### 2.2. Measurement Items and Methods

#### 2.2.1. Measurement of SPAD Values

The SPAD values of the leaves were determined using a handheld chlorophyll meter (SPAD-502Plus, konica minolta, Chiyodaku, Japan). Due to the greater physiological activity of the top leaves, they offer better observability of the plant’s physiological responses to environmental stress [20]. Therefore, the top 2 or top 3 leaves were selected for measurement while avoiding the main leaf veins. Each leaf was measured three times, and the average of these measurements was taken as the SPAD value for that particular leaf. Due to damage to individual leaves during the measurement process, a total of 85 samples were obtained.

#### 2.2.2. Measurement of Chlorophyll Fluorescence Kinetics Characteristics

The Plant Efficiency Analyzer (Pocket PEA) produced by the British company Hansatech was used to measure the rapid chlorophyll fluorescence induction kinetics and related chlorophyll fluorescence parameters of lettuce seedling leaves. The top 2 or top 3 leaves were selected for measurement while avoiding the main leaf veins. A leaf clip was used to hold the leaf blade, allowing for 30 min of dark adaptation. Then, the leaves were illuminated with a saturated red flash at a light intensity of 3500 µmol/(m^2^·s) for 2 s, and the fluorescence signal was recorded at intervals of 10 µs. Each leaf was measured three times, and the average value was used to determine the fluorescence parameters of the lettuce seedling leaves for that treatment. Due to damage to individual leaves during the measurement process, a total of 85 samples were obtained.

The rapid chlorophyll fluorescence induction kinetics curves can be analyzed using JIP-test to obtain the following chlorophyll fluorescence parameters [21,22] (Table 1).

Using Equations (1)–(3), the values of *ψ_o_*, *φE_o_*, and *φD_o_* can be calculated.
(1)$$\Psi_{o}=1-V_{j}$$
(2)$$\phi E_{o}=[1-(F_{o}/F_{m})]\times \Psi_{o}$$
(3)$$\phi D_{o}=1-\phi P_{o}=F_{o}/F_{m}$$

Equations (4) and (5) can be used to standardize the original OJIP curve to the O-P and O-J phases.
(4)$$V_{o-p}=(F_{t}-F_{o})/(F_{m}-F_{o})$$
(5)$$V_{o-J}=(F_{t}-F_{o})/(F_{J}-F_{o})$$

#### 2.2.3. Statistical Analysis Methods

The related fluorescence parameters and the fluorescence intensity over time were exported using PEA plus V1.10. Data analysis was performed using Excel 2010 for statistical calculations, Origin 2021 for graph plotting and correlation analysis, and SPSS 23.0 for significance analysis and regression analysis.

The significance test was utilized to determine whether there were significant differences in the mean SPAD values and chlorophyll fluorescence parameter means among different groups. The degree of correlation between SPAD values and chlorophyll fluorescence parameters was measured using the Pearson correlation coefficient method to identify the most strongly correlated characteristic parameters. Regression models can be employed for prediction, correlation analysis, and causal inference. Hence, in this study, the relationship between leaf SPAD values and chlorophyll fluorescence parameters was investigated using a regression model. In this context, a better fit of the model is indicated by a larger F-value, suggesting a stronger explanatory power of the model for the dependent variable. A value of R-squared closer to 1 indicates a higher degree of fit of the regression model to the observed values, signifying a closer correlation between the two variables.

## 3. Results and Discussion

### 3.1. Effects of Cadmium Stress on the SPAD Values of Lettuce Seedling Leaves

Chlorophyll is the pigment used by plants to undergo photosynthesis, and to some extent, its magnitude effectively reflects the level of photosynthetic activity. Under heavy metal stress, the chlorophyll content in plants typically decreases, thereby impacting the photosynthetic activity of the plant. Figure 1 shows the variation in SPAD values of lettuce leaves at different cadmium concentrations during the seedling stage. During the seedling stage, the maximum SPAD value of lettuce leaves was observed under cadmium stress at 5 mg/kg. Under cadmium stress at 1 mg/kg and 5 mg/kg, the SPAD values of lettuce seedling leaves were higher than those of the CK group; under cadmium stress at 10 mg/kg and 20 mg/kg, the SPAD values of lettuce seedling leaves were lower than those of the CK group. After 1 day of cadmium stress, the changes varied, and the SPAD values of lettuce leaves under different cadmium concentrations were all higher than those of the CK group. It is possible that short-term cadmium stress can enhance or accelerate certain physiological and biochemical responses in plants, leading to an increase in chlorophyll content [23]. It has been observed that 1 mg/kg and 5 mg/kg cadmium stress can promote an increase in chlorophyll content in lettuce leaves, while 10 mg/kg and 20 mg/kg cadmium stress can inhibit the increase. Under the stress of 1 mg/kg and 5 mg/kg cadmium, it is possible that metal ions can promote the activity of cytokinin metabolism enzymes, thus stimulating cell growth and leading to an increase in chlorophyll content. Under the stress of 10 mg/kg and 20 mg/kg cadmium, the expression of key enzymes in the chlorophyll synthesis pathway is inhibited, leading to a decrease in chlorophyll content [24,25]. As the duration of stress prolonged, there were no significant changes in the SPAD values of lettuce seedling leaves under 1 mg/kg and 5 mg/kg cadmium stress. However, under 10 mg/kg and 20 mg/kg cadmium stress, the SPAD values of lettuce seedling leaves gradually decreased. Among them, only 1 mg/kg and 10 mg/kg cadmium stress showed significant differences from the CK group on the first day. This may be because 1 mg/kg cadmium stress has a weaker effect on chlorophyll synthesis in lettuce leaves, and lettuce gradually develops a certain degree of adaptation to cadmium stress by consuming more energy or cooperating with other antioxidant enzymes to maintain physiological metabolic balance. Under 10 mg/kg cadmium stress, lettuce may adapt or reduce the impact of cadmium stress by regulating physiological characteristics such as net photosynthetic rate (Pn) and stomatal conductance (Gs), enhancing its self-regulation mechanism [26]. The above phenomenon indicates that low concentrations of cadmium have a promoting effect on chlorophyll synthesis, while high concentrations of cadmium have an inhibitory effect on chlorophyll synthesis. Although nitrate can also promote chlorophyll synthesis [27,28], Jia et al. [29] reached the same conclusion using cadmium chloride. Therefore, it can be considered that cadmium plays a dominant role in affecting chlorophyll synthesis.

### 3.2. Effects of Cadmium Stress on the Chlorophyll Fluorescence Parameters of Lettuce Seedling Leaves

Chlorophyll fluorescence parameters can effectively reflect the processes of energy absorption, utilization, transfer, and dissipation within the plant’s photosystem II reaction center. Therefore, the use of chlorophyll fluorescence analysis techniques can effectively help understand the damage to the photosynthetic organs of plants under adverse conditions [30].

#### 3.2.1. Effects of Cadmium Stress on the F_v_/F_m_ and PIabs of Lettuce Seedling Leaves

F_v_/F_m_ is commonly used as an indicator when analyzing chlorophyll fluorescence parameters, and under normal circumstances, its value remains stable [31]. During the seedling stage, there is a consistent decreasing trend in F_v_/F_m_ with the increase in cadmium concentration (Figure 2a). Under 20 mg/kg cadmium stress, the F_v_/F_m_ of lettuce seedling leaves showed a significant decrease compared to the control group (*p* < 0.05). Specifically, at 1, 3, 5, 7, and 9 days after cadmium stress, it decreased by 5.0%, 5.77%, 6.25%, 7.80%, and 13.48%, respectively, and with the extension of cadmium stress’ duration, F_v_/F_m_ gradually decreased. This indicates that within a specific time frame and at a certain concentration of cadmium stress, the F_v_/F_m_ of lettuce seedling leaves can remain stable, but it decreases as the duration of stress and cadmium concentration continue to increase.

PIabs can accurately reflect the status of the photosynthetic apparatus and is the most sensitive parameter among chlorophyll fluorescence parameters, capable of providing a significant indication of responses to stress [32]. The PIabs of the CK group can remain stable over time, but under cadmium stress, the PIabs of lettuce seedlings is significantly decreased compared to the CK group (*p* < 0.05) (Figure 2b), and gradually decreased with the prolongation of stress time and the increase in cadmium concentration. On the 9th day after cadmium stress, the PIabs of each group reached the lowest point. Among them, the stress of 1 mg/kg cadmium caused a significant decrease in PIabs, which was 33.69% lower than the CK group, indicating that cadmium stress can reduce the photosynthetic capacity of lettuce seedlings, decrease the activity of reaction centers, and inhibit the rate of photosynthetic electron transport [33].

#### 3.2.2. Effects of Cadmium Stress on the PSII Acceptor Side Electron Transport and Energy Allocation Ratio in Lettuce Seedling Leaves

As shown in Figure 3, V_j_ is one of the indicators reflecting the reduction situation of the central receptor in the Q_A_ test. It increases with the increase in soil cadmium concentration. The impact of cadmium stress at 10 mg/kg and 20 mg/kg on lettuce seedlings is significant. Moreover, as the stress time prolongs, the V_j_ of lettuce seedlings in each group also gradually increases. Under the cadmium concentration of 20 mg/kg, V_j_ reaches its maximum after 9 days of stress, increasing by 105.92% compared to the CK group and by 88.99% compared to the first day. Under cadmium stress, neither N nor Sm exhibited a clear pattern of change, but their values were consistently lower than those of the CK group. After 7 days of exposure to 20 mg/kg cadmium, their values reached their lowest point, showing reductions of 35.28% and 38.80%, respectively, compared to the CK group. Both *ψ_o_* and *φE_o_* decreased with the prolongation of stress time and the increase in cadmium concentration. Under 1 mg/kg cadmium stress, *ψ_o_* was significantly reduced compared to the CK group only after 7 days of stress, decreasing by 17.31%. On the other hand, *φE_o_* was noticeably reduced compared to the CK group after 5 days of cadmium stress, with a reduction of 11.58%. The *φD_o_* increased with the prolongation of stress time and the increase in cadmium concentration. Under 1 mg/kg and 5 mg/kg cadmium stress, it significantly increased compared to the CK group until the 5th day, with respective increases of 13.34% and 15.41% compared to the CK group. The above phenomenon indicates that cadmium stress leads to the inability of electrons to transfer from Q_A_ to Q_B_ on the PSII, resulting in reduced electron transfer from Q_A_^−^, thereby decreasing the proportion of energy for subsequent electron transfer from Q_A_^−^. This impedes linear electron transport, causing damage to the electron transfer chain and reducing the energy entering the electron transfer in the photosynthetic system. This affects the energy distribution ratio of PSII, leading to a decrease in electron transfer efficiency [34]. Furthermore, this effect becomes more pronounced with the prolongation of stress time and the increase in soil cadmium concentration. However, within a certain time range and at a certain cadmium concentration, lettuce seedlings exhibit a certain level of resistance to cadmium stress.

#### 3.2.3. Effects of Cadmium Stress on the Activity of Photosystem II (PSII) Reaction Centers in Lettuce Seedling Leaves

According to Figure 4, both ABS/RC and DI_O_/RC showed a significant increase compared to the CK group under cadmium stress at 10 mg/kg and 20 mg/kg, and this increase gradually intensified with prolonged stress duration. Under 1 mg/kg cadmium stress, ABS/RC and DI_O_/RC both showed a significant increase compared to the CK group only after the 9th day, with an increase of 4.32% and 25.96%, respectively. However, DI_O_/RC exhibited a decrease of 6.97% compared to the CK group after the 1st day. Under 5 mg/kg cadmium stress, ABS/RC showed a significant increase compared to the CK group after the 3rd day, with an increase of 1.40%, while DI_O_/RC showed a significant increase compared to the CK group only after the 9th day, with an increase of 21.58%. TR_O_/RC exhibited a pattern of initially decreasing and then increasing under cadmium stress. After 1 day and 3 days of cadmium stress, all groups showed a decrease compared to the CK group. However, with the prolongation of stress time, TR_O_/RC under 10 mg/kg and 20 mg/kg cadmium stress significantly increased compared to the CK group. After 9 days of cadmium stress, it increased by 8.69% and 10.90%, respectively, compared to the CK group. On the other hand, under 1 mg/kg and 5 mg/kg cadmium stress, TR_O_/RC showed no significant change compared to the CK group. ET_O_/RC decreased with the increase in soil cadmium concentration, and the extent of decrease became greater with the prolongation of stress time. Specifically, under 1 mg/kg cadmium stress, ET_O_/RC was significantly reduced compared to the CK group only after 3 days of stress, with a reduction of 8.90%. The above phenomenon indicates that within a certain time frame and at a certain cadmium concentration, lettuce seedlings exhibit a certain resistance to cadmium stress. However, as the duration of stress and the cadmium concentration increase, the energy used for electron transfer in the unit reaction center of the lettuce seedlings decreases. Under 10 mg/kg and 20 mg/kg cadmium stress, although the energy absorbed and captured by the unit reaction center increases, the energy dissipated in the form of heat also increases, while the energy used for photosynthesis is minimal. As a result, excess light energy for photosynthesis is regulated through thermal dissipation to reduce the damage to the PSII reaction center, which is considered a self-protection mechanism of lettuce seedlings under adverse stress conditions [35].

### 3.3. Effects of Cadmium Stress on the Rapid Chlorophyll Fluorescence Kinetics Characteristics of Lettuce Seedling Leaves

#### 3.3.1. Effects of Cadmium Stress on the Induction Kinetics Curve of Rapid Chlorophyll Fluorescence in Lettuce Seedling Leaves

The OJIP curve can reflect a wealth of information about the primary photochemical reactions of PSII reaction centers. Through the analysis of the OJIP curve, variations in PSII electron supply, transfer, and PSII reaction center activity can be determined [36]. As shown in Figure 5, the fluorescence intensity at points O, K, and J increases with the increase in cadmium concentration in the soil. However, under 1 mg/kg cadmium stress, the changes in fluorescence intensity at these points are not significant compared to the CK group. With a further increase in cadmium concentration, significant changes were observed. Under 5 mg/kg cadmium stress, the fluorescence intensity at points O, K, and J increased by 11.09%, 19.60%, and 14.57%, respectively, compared to the CK group. The fluorescence intensity at point I did not show significant changes under cadmium stress. The fluorescence intensity at point P did not exhibit noticeable changes at cadmium concentrations of 1 mg/kg and 5 mg/kg. However, it significantly decreased at cadmium concentrations of 10 mg/kg and 20 mg/kg, with reductions of 3.81% and 4.90%, respectively, compared to the CK group. The results indicate that within a certain range, lettuce seedlings exhibit a certain degree of resistance to cadmium stress; however, as the cadmium concentration continues to increase, the photosynthesis of lettuce seedlings is inhibited.

#### 3.3.2. Standardization of the O-P and O-J Phases of the OJIP Curve

Due to significant interference from external factors, the information reflected in the original OJIP curve needs to be standardized [37]. In this study, the original OJIP curve was standardized for O-P and O-J, with the analysis focusing on the 0–2 ms segment of the image. As shown in Figure 6a,c, after standardizing the O-P of the OJIP curve, it was observed that the fluorescence intensity at points J and I increased with cadmium concentration. Under 20 mg/kg cadmium stress, the fluorescence intensity at points J and I increased by 33.30% and 2.10%, respectively, compared to the CK group. This indicates that cadmium stress can inhibit electron transfer on the PSII acceptor side of lettuce seedling leaves. The variation in relative fluorescence intensity at the K point of the OJIP curve at 0.3 ms can effectively reflect the damage to the structure and function of the oxygen-evolving complex (OEC) under stress conditions. After normalization of the O-P phase, the fluorescence intensity at the K point increased with increasing cadmium concentration. It increased by 5.01%, 34.75%, 74.68%, and 88.59%, respectively, compared to the CK group. After standardization of the O-J phase, the fluorescence intensity at the K point gradually increased with increasing cadmium concentration (ΔV_o-J_ > 0), as shown in Figure 6b,d. It increased by 9.36%, 12.14%, 36.21%, and 41.48%, respectively, compared to the CK group. Cadmium stress has been shown to induce damage to the OEC of lettuce seedlings, thereby inhibiting photolysis of water and reducing the supply of electrons from OEC to PSII. As a result, the efficiency of electron transfer is decreased, which is one of the contributing factors to the decline in photosynthetic rate.

### 3.4. Analysis of the Correlation between Chlorophyll Fluorescence Parameters and SPAD Values

To identify the chlorophyll fluorescence parameters that can estimate SPAD values, a correlation analysis was conducted between the 12 chlorophyll fluorescence parameters and SPAD values. The results are presented in Figure 7. Red color indicates positive correlation, while blue color indicates negative correlation. The size of the circle represents the degree of correlation, with larger circles indicating a higher correlation. Among the 12 chlorophyll fluorescence parameters, ABS/RC, DI_O_/RC, TR_O_/RC, and ET_O_/RC exhibit a strong correlation with SPAD values. The research findings indicate that under cadmium stress, there is a significant association between the unit PSII reaction center activity and SPAD values in lettuce leaves. This could be attributed to the increased oxidative stress caused by cadmium stress, which leads to an excessive production of reactive oxygen species and damages the photosynthetic system, resulting in a decline in the activity of the PSII reaction center and subsequently impacting chlorophyll synthesis and stability [38,39]. Li et al. [40] found that there is a high correlation between the V_j_ chlorophyll fluorescence parameter and chlorophyll content in soybeans during the grain fermentation period. This discrepancy may be caused by differences in the test materials and the conditions of stress.

### 3.5. Establishment of a SPAD Value Estimation Model for Lettuce Leaves under Cadmium Stress

The average values of the sample data for each cadmium stress treatment at different time points were taken as the modeling sample data, totaling 25 samples. The chlorophyll fluorescence parameters ABS/RC, DI_O_/RC, TR_O_/RC, and ET_O_/RC were chosen as independent variables, while the SPAD value was selected as the dependent variable. Regression models were established using linear, quadratic, cubic, and logarithmic functions, and the results are shown in Table 2. Among them, the model based on ET_O_/RC showed the best fit with the SPAD values of lettuce leaves under cadmium stress, with a coefficient of determination (R^2^) of 0.684. Yang et al. [41] suggested that the number of active reaction centers per unit area can be used to estimate chlorophyll content under non-stress conditions, which differs from the conclusions of this study and may be attributed to cadmium stress.

### 3.6. Model Validation

To further examine the accuracy of the chlorophyll estimation model and prevent overfitting, 30% of the sample data were selected for model validation [42]. The chlorophyll fluorescence parameters ABS/RC, DI_O_/RC, TR_O_/RC, and ET_O_/RC were used as independent variables to predict SPAD values, which were then compared with the actual measurements. As shown in Figure 8, after model validation, it was found that among the four models, DIO/RC and ETO/RC showed better estimation performance for chlorophyll content. Specifically, the model built with ETO/RC demonstrated the best estimation performance, with an R^2^ value of 0.796.

## 4. Conclusions

This study focused on lettuce seedlings and examined the SPAD values, chlorophyll fluorescence parameters, and chlorophyll fluorescence kinetics of lettuce seedling leaves under cadmium stress using exogenously added cadmium in pot planting. The following conclusions were drawn:(1)Cadmium stress with concentrations of 1 mg/kg and 5 mg/kg has a promoting effect on the SPAD value of lettuce seedling leaves, while cadmium stress concentrations of 10 mg/kg and 20 mg/kg have an inhibitory effect on the SPAD value of lettuce seedling leaves. Moreover, with prolonged exposure time to stress, the inhibitory effect became more pronounced.(2)Cadmium stress affects both the donor and acceptor sides of PSII in lettuce seedling leaves, damaging the electron transport chain and reducing the energy entering the electron transfer in the photosynthetic system. Additionally, it also damages the OEC, inhibiting water photolysis and reducing the electrons supplied by OEC to PSII, resulting in a decrease in electron transfer efficiency and a decline in photosynthetic rate. However, lettuce seedling leaves can mitigate the damage caused by cadmium stress to the PSII reaction center by increasing thermal dissipation.(3)The model based on ET_O_/RC effectively predicts the SPAD values of lettuce seedling leaves and can be considered an effective method for estimating leaf chlorophyll content under cadmium stress.

The chlorophyll fluorescence analysis technique can not only be used for studying the physiological mechanisms of lettuce under stress conditions but also allows for the simultaneous estimation of leaf chlorophyll content. This approach effectively simplifies the experimental process and enables the synchronized determination and correlation analyses between chlorophyll content and other physiological indicators in lettuce leaves.

## Figures and Tables

**Figure 1 sensors-24-01501-f001:**
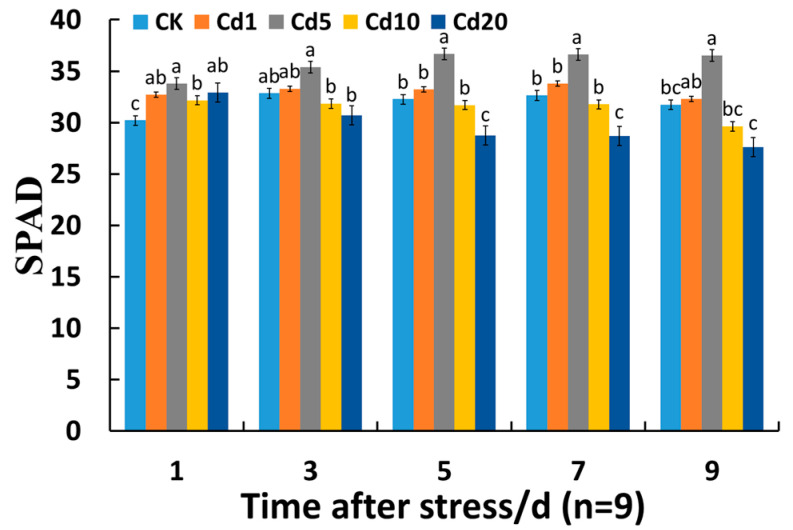
Effects of cadmium stress on the SPAD value of lettuce seedling leaves. Different lowercase letters indicate significant differences at the *p* < 0.05 level.

**Figure 2 sensors-24-01501-f002:**
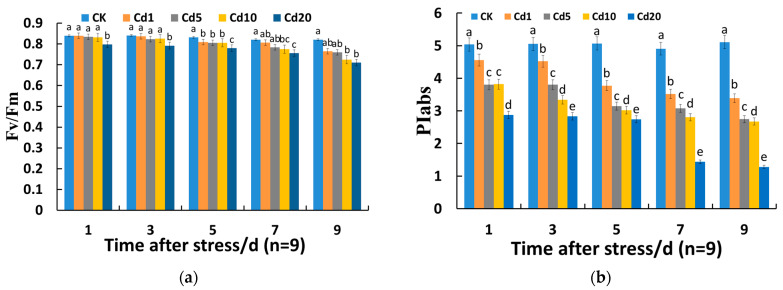
Effects of cadmium stress on chlorophyll fluorescence parameters in lettuce seedlings: (**a**) effects of cadmium stress on the *F_v_/F_m_* of lettuce seedling leaves; (**b**) effects of cadmium stress on the PIabs of lettuce seedling leaves. Different lowercase letters indicate significant differences at the *p* < 0.05 level.

**Figure 3 sensors-24-01501-f003:**
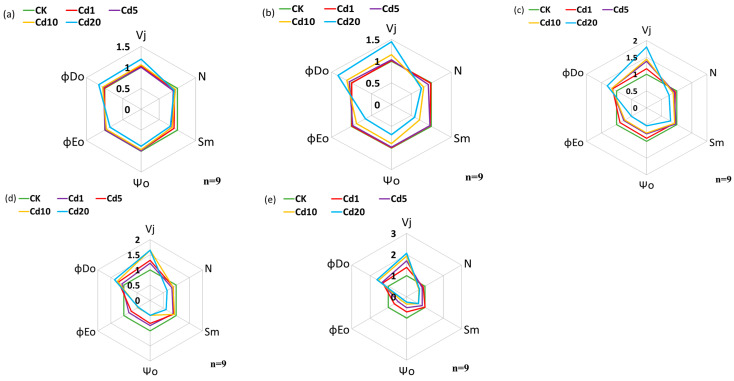
Effects of cadmium stress on chlorophyll fluorescence parameters in lettuce seedlings: (**a**) 1 day after cadmium stress; (**b**) 3 days after cadmium stress; (**c**) 5 days after cadmium stress; (**d**) 7 days after cadmium stress; (**e**) 9 days after cadmium stress.

**Figure 4 sensors-24-01501-f004:**
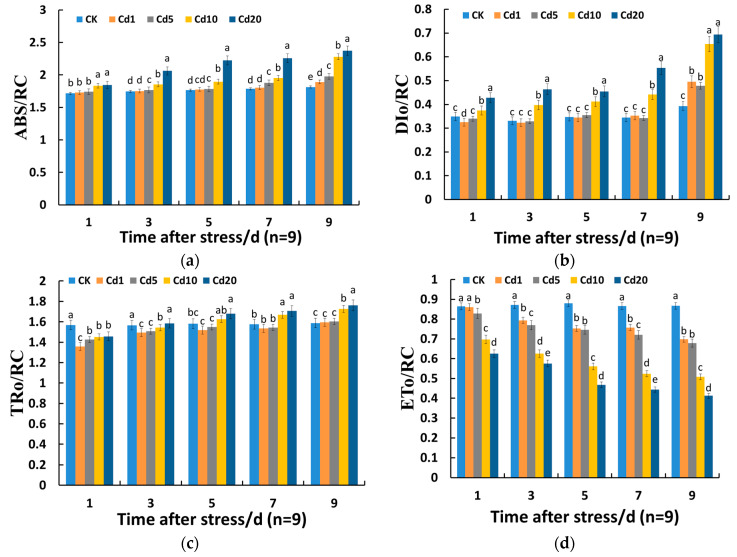
Effects of cadmium stress on the activity of photosystem II reaction centers in lettuce seedling leaves: (**a**) effects of cadmium stress on the ABS/RC of lettuce seedling leaves; (**b**) effects of cadmium stress on the DI_O_/RC of lettuce seedling leaves; (**c**) effects of cadmium stress on the TR_O_/RC of lettuce seedling leaves; (**d**) effects of cadmium stress on the ET_O_/RC of lettuce seedling leaves. Different lowercase letters indicate significant differences at the *p* < 0.05 level.

**Figure 5 sensors-24-01501-f005:**
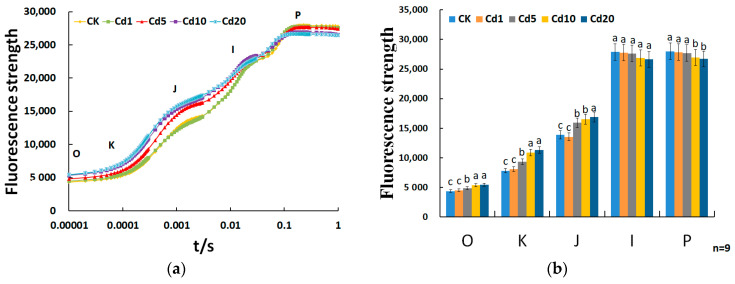
OJIP curves of lettuce seedlings leaves under cadmium stress: (**a**) trends in OJIP curve variation; (**b**) differential analysis of points O, J, I, and P. Different lowercase letters indicate significant differences at the *p* < 0.05 level.

**Figure 6 sensors-24-01501-f006:**
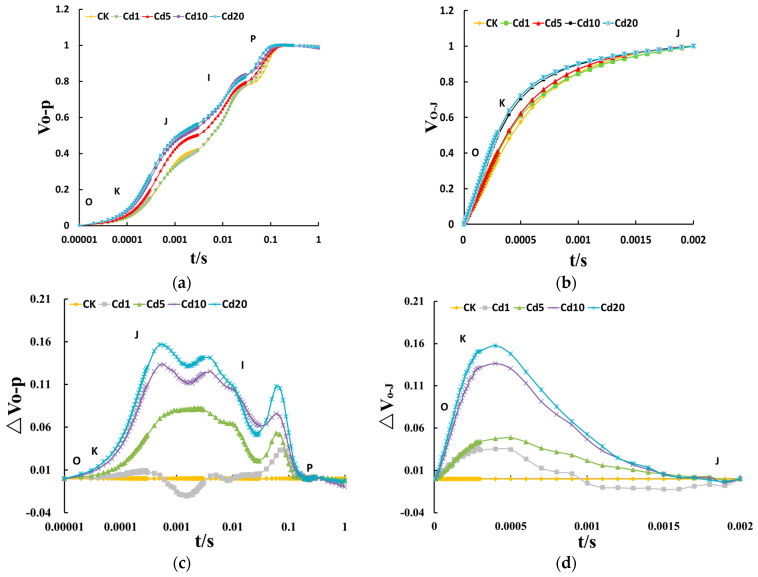
Standardized O-P and O-J curves and relative variable fluorescence difference analysis of lettuce seedling leaves under cadmium stress: (**a**) standardized fluorescence curve at O-P point; (**b**) standardized fluorescence curve at O-J point; (**c**) normalized differential fluorescence curve between O-P point and control group data; (**d**) normalized differential fluorescence curve between O-J point and control group data.

**Figure 7 sensors-24-01501-f007:**
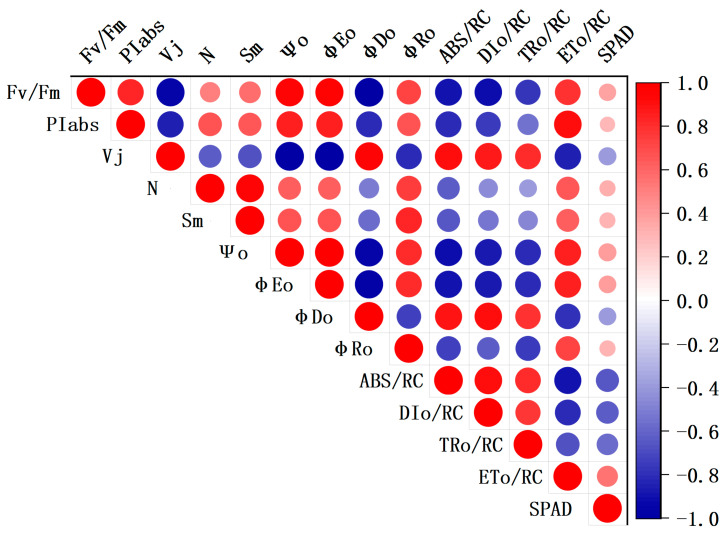
Correlation between chlorophyll fluorescence parameters and SPAD values under cadmium stress.

**Figure 8 sensors-24-01501-f008:**
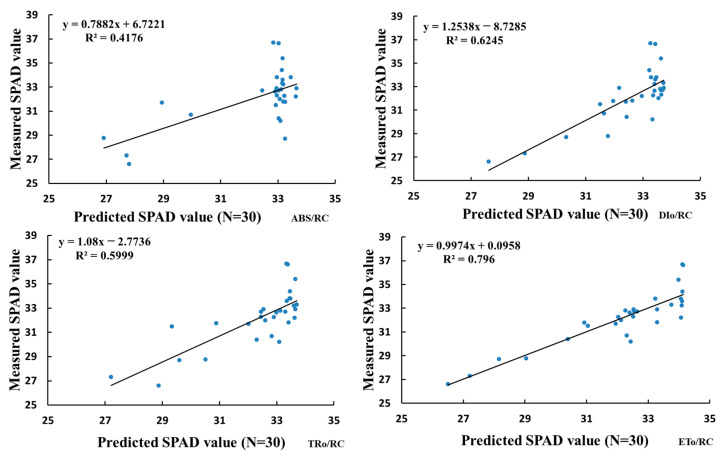
Fitted curve of predicted values and actual measurements.

**Table 1 sensors-24-01501-t001:** Parameter description.

Parameter	Parameter Description
*F_o_*	Initial fluorescence
*F_m_*	Maximum fluorescence
*F_v_*	Variable fluorescence
*F_t_*	The variable fluorescence intensity at each point
*F_v_/F_m_*	The maximum quantum yield of photosystem II
PIabs	The performance index based on absorbed light energy
V_j_	The relative variable fluorescence intensity at the J point
N	The number of oxidized–reduced turnovers of Q_A_
S_m_	The PSII acceptor side electron transfer complex
*ψ_o_*	The probability of captured excitons transferring electrons into the electron transport chain
*ϕE_o_*	The quantum yield of electron transport
*ϕD_o_*	The quantum ratio used for thermal dissipation
*ϕR_o_*	The quantum yield for reduction of the PSI acceptor side end electron acceptors
ABS/RC	The light energy absorbed per reaction center
DI_o_/RC	The heat dissipation per reaction center
TR_o_/RC	The energy captured by a reaction center for reducing Q_A_
ET_o_/RC	The energy captured by a reaction center for electron transfer
Q_A_	Primary bound plastoquinone
Q_B_	Secondary bound plastoquinone
O	Minimal fluorescence
K	Fluorescence at 0.3 ms
J	Fluorescence at 2 ms
I	Fluorescence at 30 ms
P	Peak fluorescence

**Table 2 sensors-24-01501-t002:** SPAD value estimation models.

Parameters	Fitting Model	R^2^	Significance	F	Standard Error
ABS/RC	$y=-42.933+82.715x-22.451x^{2}$	0.503	**	11.135	1.746
DI_O_/RC	y=38.476−14.734x	0.386	**	14.442	1.898
TR_O_/RC	$y=-147.093+243.97x-82.301x^{2}$	0.502	**	11.107	1.747
ET_O_/RC	$y=3.95+62.238x-39.181x^{3}$	0.684	**	23.818	1.392

Note: ** indicates significant correlation at the 0.01 level.

## Data Availability

The data that support the findings of this study are available upon reasonable request from the authors.

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
