# Peer review of "Application of Chlorophyll Fluorescence Analysis Technique in Studying the Response of Lettuce (Lactuca sativa L.) to Cadmium Stress"

_sensors, 2024, doi:10.3390/s24051501_

Round 1
Reviewer 1 Report
Comments and Suggestions for Authors
The manuscript by Zhou et al. is devoted to the study the mechanisms of influence of cadmium stress on lettuce seedlings. It is an important problem, and this work contains very interesting results, but I have some comments to the manuscript:
1. P. 3, lines 107-108: “To ensure the normal growth of lettuce seedlings throughout the entire experiment, ensure an ample water supply.” – It is necessary to provide quantitative parameters of water availability (soil moisture or at least frequency of watering) in the text.
2. P. 3, Table 1: About Fv/Fm: “The photosystem II quantum yield” – It is better: “maximum quantum yield of photosystem II”.
3. P. 5, lines 185-186: “At the concentration of 1mg/kg and 5mg/kg, cadmium stress can promote the increase in chlorophyll content in lettuce leaves.”. SPAD at 1 mg/kg differs significantly from the control only on the 1st day (as at other concentrations on the 1st day).
4. P. 5, lines 186-188: “However, at the concentrations of 10mg/kg and 20mg/kg, cadmium stress inhibits the increase in chlorophyll content in lettuce leaves.”. There was no significant difference on the days 3, 5, 7, 9 for 10 mg/kg.
5. All figures: The number of data points (n) used to calculate the means and SEM should be added to the legends.
6. P. 13, lines 413-414: “Under cadmium stress, the SPAD values of lettuce seedling leaves showed inhibition at high concentrations and promotion at low concentrations.” By what criterion did you classify concentrations into high and low? What is the reason for choosing these particular concentrations? I think this needs to be noted.
Author Response
Dear Editor and Reviewers,
Thank you for your detailed and important comments on our manuscript. After receiving the review comments, all eight authors of the manuscript have carefully analyzed and discussed the review comments. We believe that these review comments are very constructive and have therefore completed the revisions one by one.
The specific revisions are described below (review comments are in red, responses and revisions are in black):
Comments 1: P. 3, lines 107-108: “To ensure the normal growth of lettuce seedlings throughout the entire experiment, ensure an ample water supply.” – It is necessary to provide quantitative parameters of water availability (soil moisture or at least frequency of watering) in the text.
Response 1: Thank you for pointing this out. We deeply apologise for this, due to our negligence, your understanding of the article has brought you a bad experience. We agree with this comment. Therefore, we provided quantitative parameters of water availability in the experimental design. Mention exactly where in the revised manuscript this change can be found – page number 3, paragraph 1, and line 110.
Comments 2: P. 3, Table 1: About Fv/Fm: “The photosystem II quantum yield” – It is better: “maximum quantum yield of photosystem II”.
Response 2: Agree. We have, accordingly, made modifications in Table 1 to emphasize this point. Mention exactly where in the revised manuscript this change can be found – page number, paragraph 4, and line 139.
Comments 3: P. 5, lines 185-186: “At the concentration of 1mg/kg and 5mg/kg, cadmium stress can promote the increase in chlorophyll content in lettuce leaves.”. SPAD at 1 mg/kg differs significantly from the control only on the 1st day (as at other concentrations on the 1st day).
Response 3: Thank you for raising this issue. We have taken note of this issue. We have provided a detailed explanation of the phenomenon that "SPAD at 1 mg/kg showed a significant difference from the control group only on the first day". Additionally, we have repositioned the original interpretation in the text. Mention exactly where in the revised manuscript this change can be found – page number 5, and line 185-188 and 195-198.
Comments 4: P. 5, lines 186-188: “However, at the concentrations of 10mg/kg and 20mg/kg, cadmium stress inhibits the increase in chlorophyll content in lettuce leaves.”. There was no significant difference on the days 3, 5, 7, 9 for 10 mg/kg.
Response 4: Agree. Thank you for your feedback. We apologize deeply for our oversight. We have provided an explanation for the phenomenon that "10 mg/kg showed no significant difference on the 3rd, 5th, 7th, and 9th days". Mention exactly where in the revised manuscript this change can be found – page number, paragraph 5, and line 198-201.
Comments 5: All figures: The number of data points (n) used to calculate the means and SEM should be added to the legends.
Response 5: Agreed. We appreciate your valuable suggestions and apologize for any figures issues in the article. We have thoroughly examined the manuscript and made necessary adjustments to the article, including replacing the images. Mention exactly where in the revised manuscript this change can be found – page number, paragraph 6-10 and 13, and line 207,243,280,315,343 and 418.
Comments 6: P. 13, lines 413-414: “Under cadmium stress, the SPAD values of lettuce seedling leaves showed inhibition at high concentrations and promotion at low concentrations.” By what criterion did you classify concentrations into high and low? What is the reason for choosing these particular concentrations? I think this needs to be noted.
Response 6: Agree. Thank you for raising this issue. We have taken note of this issue. We have stopped classifying the concentration as high and low. Instead, we have replaced the high concentration with 10 mg/kg and 20 mg/kg, and the low concentration with 1 mg/kg and 5 mg/kg to emphasize this point. The principle for selecting these specific concentrations is based on several factors. According to the "Environmental Quality Standards for Soils" (GB15618-2008), the second-tier quality standard for total cadmium in agricultural land is 1.0 mg/kg, with a limit of 0.6 mg/kg for vegetable plots with pH > 7.5. The "Maximum Levels of Contaminants in Foods" (GB2762-2012) sets the cadmium limit in leafy vegetables at 0.2 mg/kg. Based on addition and recovery experiment guidelines, soil cadmium levels were set at 1 mg/kg and 5 mg/kg, and considering a survey showing a range of 0.2 to 20.0 mg/kg for cadmium in agricultural soils, concentrations of 0 (control), 1, 5, 10, and 20 mg/kg were chosen. Mention exactly where in the revised manuscript this change can be found – page number, paragraph 13, and line 426-428.

Reviewer 2 Report
Comments and Suggestions for Authors
This study investigates the impact of cadmium stress on lettuce seedlings by analyzing chlorophyll fluorescence. It shows that lower cadmium concentrations promote chlorophyll content while higher concentrations inhibit it. This study also proposed a model for predicting the SPAD value.
1. For section 3.5 and 3.6, 4 models were proposed to predict SPAD value. 25 samples were taken as sample data. While in section 3.6, 10 random samples were selected from 85 samples. Why do you not use those 85+25-10=100 samples for verification purposes? It would be better if you could give more descriptions of those samples or simply provide them in the SI.
2. It seems all 4 models perform much better in figure 8 than in the sample data set (table 2), if you compare those R2 values. Could you give more explanation?
3. Page 2 line 60 and others, please confirm if it is a specific technique “chlorophyll fluorescence technique” because this doesn’t sound like a technique. Usually, it’s just the analysis of the chlorophyll fluorescence signal/spectrum.
4. Please check section 2.1 materials and experimental design. It should mostly be past tense.
5. Page 5, line 174, this “seedling stage” is a little confusing, cause readers may think day1-9 are all seedling stages. However, it seems that “V” trend only exists in day 1.
6. Page 6, line 227, why is 1 mg/kg cadmium result brought out for discussion? It seems all concentrations of cadmium caused significant decrease in PIabs. Also, please make “PIabs” uniform, either all subscript or not.
Comments on the Quality of English Languageminor edits in materials and experimental design are needed
Author Response
Dear Editor and Reviewers,
Thank you for your detailed and important comments on our manuscript. After receiving the review comments, all eight authors of the manuscript have carefully analyzed and discussed the review comments. We believe that these review comments are very constructive and have therefore completed the revisions one by one.
The specific revisions are described below (review comments are in red, responses and revisions are in black):
Comments 1: For section 3.5 and 3.6, 4 models were proposed to predict SPAD value. 25 samples were taken as sample data. While in section 3.6, 10 random samples were selected from 85 samples. Why do you not use those 85+25-10=100 samples for verification purposes? It would be better if you could give more descriptions of those samples or simply provide them in the SI.
Response 1: Thank you for pointing this out. We deeply apologise for this, due to our negligence, your understanding of the article has brought you a bad experience. We agree with this comment. Therefore, we have provided a new interpretation of this section to emphasize this point. The reason for not using all 85+25-10=100 samples for validation is primarily due to the practical constraints of limited resources and time in actual research. Processing and analyzing a large number of samples require more time and computational resources. Therefore, based on experimental design needs and available resources, a deliberate decision was made to select a smaller number of samples for model validation. However, validating with a smaller sample size or a larger sample size can both potentially lead to overfitting. To prevent overfitting, approximately 30% of the samples were selected for validation in the revised manuscript. Mention exactly where in the revised manuscript this change can be found – page number 12, paragraph 2, and line 410-411.
Comments 2: It seems all 4 models perform much better in figure 8 than in the sample data set (table 2), if you compare those R2 values. Could you give more explanation?
Response 2: Agree. Thank you for raising this issue. We have taken note of this issue. We carefully examined the content of this section and provided a revised explanation. The reason for this phenomenon may be attributed to the limited number of samples used for validation, leading to overfitting. Therefore, increasing the number of validation samples can help mitigate this issue. Mention exactly where in the revised manuscript this change can be found – page number12-13, and line 411-417.
Comments 3: Page 2 line 60 and others, please confirm if it is a specific technique “chlorophyll fluorescence technique” because this doesn’t sound like a technique. Usually, it’s just the analysis of the chlorophyll fluorescence signal/spectrum.
Response 3: Agreed. We appreciate your valuable suggestions and apologize for any conceptual issues in the article. We have defined this concept. The "chlorophyll fluorescence technique" is indeed a specific method widely utilized in plant physiology and ecological studies. This technique capitalizes on the property of chlorophyll molecules to emit fluorescence after absorbing light energy. By measuring parameters of chlorophyll fluorescence, it is possible to non-destructively assess aspects such as photosynthetic efficiency, photoprotective mechanisms, and a plant's response to environmental stress.
Comments 4: Please check section 2.1 materials and experimental design. It should mostly be past tense.
Response 4: Agreed. We deeply apologize for any grammar issues in our article. We have thoroughly reviewed the entire manuscript and made necessary revisions, particularly in the Materials and Methods section, regarding the experimental design. Mention exactly where in the revised manuscript this change can be found – page number2-3, and line 91-115.
Comments 5: Page 5, line 174, this “seedling stage” is a little confusing, cause readers may think day1-9 are all seedling stages. However, it seems that “V” trend only exists in day 1.
Response 5: Agree. Thank you for your feedback. We apologize deeply for our oversight. We have rephrased that section. Mention exactly where in the revised manuscript this change can be found – page number 5, and line 175-176.
Comments 6: Page 6, line 227, why is 1 mg/kg cadmium result brought out for discussion? It seems all concentrations of cadmium caused significant decrease in PIabs. Also, please make “PIabs” uniform, either all subscript or not.
Response 6: Agree. Thank you for your feedback. The reason for discussing the results of 1 mg/kg cadmium is because it was the lowest concentration used in this study (excluding the control group). Analyzing the impact of 1 mg/kg on PIabs was done to demonstrate that even the lowest concentration had a significant effect on PIabs. This suggests that at higher concentrations, cadmium stress would have a more pronounced impact on lettuce. Additionally, the revised manuscript has standardized the term "PIabs" throughout the article. Mention exactly where in the revised manuscript this change can be found – page number 4 and 6, and line 139,219 and 231-241.
Point 1: minor edits in materials and experimental design are needed
Response 1: Agreed. Thank you for bringing this to our attention. According to your valuable suggestion, we thoroughly reviewed the article and made extensive use of the past tense in the Materials and Methods section for most of the content. Mention exactly where in the revised manuscript this change can be found – page number2-3, and line 91-115.
